# Genetic Contributions to Aggressive Behaviour in Pigs: A Comprehensive Review

**DOI:** 10.3390/genes16050534

**Published:** 2025-04-29

**Authors:** Anastasiya Kazantseva, Airat Bilyalov, Nikita Filatov, Stepan Perepechenov, Oleg Gusev

**Affiliations:** 1Institute of Biochemistry and Genetics, Ufa Federal Research Centre of Russian Academy of Sciences, 450054 Ufa, Russiabilyalovair@yandex.ru (A.B.);; 2SBHI Moscow Clinical Scientific Center Named after Loginov MHD, 111123 Moscow, Russia; 3Life Improvement by Future Technologies (LIFT) Center, 121205 Moscow, Russia; 4Institute of Fundamental Medicine and Biology, Kazan Federal University, 420008 Kazan, Russia; 5Intractable Disease Research Center, Graduate School of Medicine, Juntendo University, Tokyo 113-8421, Japan

**Keywords:** pig aggression, genetic markers of aggression, animal welfare, genome-wide association studies

## Abstract

Aggressive behaviour in pigs poses significant challenges to animal welfare, production efficiency, and economic performance in the pork industry. This review explores the multifaceted causes of pig aggression, focusing on genetic, environmental, and physiological factors. Aggression in pigs is categorized into social, maternal, fear-induced, play, and redirected aggression, with early-life hierarchies and environmental stressors playing critical roles. Physiological markers, such as elevated cortisol and reduced serotonin levels, are closely linked to aggressive behaviour, while dietary interventions, including tryptophan supplementation, have shown promise in mitigating aggression. Environmental factors, such as overcrowding, noise, and heat stress, exacerbate aggressive tendencies, whereas enrichment strategies, like music and improved housing conditions, can reduce stress and aggression. Genome-wide analyses have pinpointed specific polymorphisms in neurotransmitter genes (*DRD2*, *SLC6A4*, *MAOA*) and stress-response loci (*RYR1*) as significant predictors of porcine aggression. Advances in genomic technologies, including genome-wide association studies (GWASs) and transcriptomic analyses, have further elucidated the genetic and epigenetic underpinnings of aggressive behaviour. Practical application in breeding programmes remains challenging due to aggression polygenic nature and industry hesitancy toward genomic approaches. Future research should focus on integrating genetic markers into breeding programmes, developing multitrait selection indices, and exploring epigenetic modifications to improve animal welfare and production efficiency. By addressing these challenges, the pork industry can enhance both the well-being of pigs and the sustainability of production systems.

## 1. Introduction


**
*The significance of pig production in agriculture and related disciplines:*
**


Pork production is a socially and economically significant component of modern society, providing raw materials for a wide range of consumer products. It is the second largest in the world in terms of volume, surpassed only by poultry meat production, and accounts for 34.23% of total meat production [1]. In addition to their importance in the domain of food production, pigs have a substantial presence in biomedical research. Their anatomical, histological, and topographical similarity to humans render them suitable for use in clinical research, including the modelling of trauma and acute conditions, as well as basic science. Pigs also serve as cadaveric material for the development of basic and experimental surgical and biomedical concepts [2,3,4,5].

## 2. The Issue of Pig Aggression and Its Repercussions on the Efficacy of Agricultural Production

Pig aggression is a significant problem in the pig industry, impacting animal welfare, productivity, and economic performance [6]. Studies indicate that aggression is common when pigs are mixed, with a notable occurrence of aggressive behaviours leading to stress and injuries among the animals. However, specific statistical data on the overall prevalence of aggression in pig populations are limited [7]. One study noted that aggression is rare in stable social groups but can escalate to lethal gang aggression in certain conditions, with 22% of victims surviving due to timely farmer intervention, although most attacks result in death. This review summarizes the current state of knowledge on the classification of pig aggression, its primary causes, and the potential role of molecular and genetic markers in its manifestation. Despite a significant scale of pig production, accumulated historical experience, and modern scientific advances in the field of genetics and breeding, animal aggression remains a substantial and considerable economic problem in the context of livestock production [8].

It is an irrefutable fact that elevated levels of aggression inevitably engender elevated levels of stress, which exert a deleterious effect on the health and well-being of pigs, with the resultant consequences of reduced feed intake, lower growth rates, increased susceptibility to disease, and diminished overall productivity. The economic benefits of studying pig behaviour and aggression are evident. The implementation of effective strategies to reduce aggression has been shown to improve the overall health and well-being of the herd, leading to economic and research benefits.

### Definition, Detection, and Classification of Pig Aggressiveness

As with other placental mammals, the pig has a well-developed psyche and displays complex social features [9]. One aspect of this social behaviour is pig aggression, a complex behaviour that can be divided into several categories depending on its underlying causes and consequences. A comprehensive understanding of the etiology and taxonomy of pig aggression is imperative for the development of effective correction strategies.

The manifestation of aggressive behaviour in pigs is observed within the context of a group and is influenced by a multifaceted set of factors. These include, but are not limited to, dominant behaviour, as well as external environmental influences such as housing conditions, the nature of the field (open or closed), the diet, and the timing of weaning [9,10].

Aggression in pigs can be broadly categorized into the following types [11,12,13]:(1)Social aggression is defined as a form of aggression that occurs within a social framework. It is frequently associated with the establishment of a dominance hierarchy between individuals. It has been particularly prevalent in the context of pig interactions, particularly within group housing systems. The manifestations of social aggression may include physical confrontations such as fighting, biting, or pushing.(2)Maternal aggression is considered as the display of behaviour directed toward the protection of offspring from potential threats. This behaviour is critical for the survival of the piglets and may include aggressive postures, loud cries, and physical attacks on other individuals or humans.(3)The phenomenon of fear-induced aggression has been observed in pigs, with evidence indicating that these animals exhibit aggressive behaviour in conditions and situations associated with fear caused by threats from other individuals or humans or from stressors such as loud noises and the presence of unfamiliar animals.(4)Aggressive behaviour has been reported in young piglets during non-stressful interactions, such as play, where imitations of confrontations and pursuits have been noted. The significance of this behaviour lies in its role in the development of social skills and the understanding of social interactions.(5)Redirected aggression is defined as a variant of aggressive behaviour, in which a pig fails to express aggression towards an intended target and instead redirects it towards another pig or object. This phenomenon has been observed in contexts involving confined animals or those exposed to stressful situations.

The manifestation of aggression in piglets can be determined from the first week of life. During this period, the feeding order is established. The introduction of new piglets into an existing group can also cause aggressive behaviour, which can be attributed to alterations in the established social hierarchy [14]. Due to the fact that sows give birth at different times, a new problem arises—aggression at further mixing of different groups of piglets. In the absence of specific conditions, a new struggle for a place in the hierarchy begins. In more natural conditions for pigs, i.e., in small groups, conflicts between different groups of piglets from different sows do not occur, including those among piglets and older sows [15].

Establishing a hierarchical structure in a new group of mixed piglets is a process that takes several days to complete. Once established, this hierarchy persists for the duration of the group’s existence, potentially extending up to its complete lifespan. Conflicts within established groups are rarely initiated, typically without unprovoked displays of aggression. In such cases, strong displays of aggression, such as tail and body biting, are more frequently associated with deteriorating conditions on the farm than with the establishment of a new order in the existing hierarchy [16].

It is evident that the early detection and isolation of aggression, in addition to the identification of patterns of aggressive behaviour, constitutes a pivotal step in the process of correcting aggression in pigs.

Observation and detection can be performed by an external observer. These are conventional observation methods based on the assessment of aggression under different conditions. These methods encompass the monitoring of social interactions during feeding, group interactions, and observations in altered animal housing conditions. The other method, which is based on automated detection, has only come into use in the last few years [17]. The method is predicated on the observation and detection of individual behaviour patterns using computer vision technology and deep machine learning. This methodology facilitates the tracking of individual pigs and the subsequent analysis of their movements. Such a system is able to provide a more complete data set over a longer period of time, without distortion due to subjective perception of the observer [18].

Automated recognition systems demonstrate effectiveness in detecting aggressive behaviour, despite limited interpretability of their artificial intelligence models [18]. How-ever, these systems may face significant limitations: their performance in dense herds and dynamic lighting conditions remains unverified, and the substantial infrastructure in-vestments required for high-performance systems may prove economically prohibitive for medium-sized farms.

The result of aggressive behaviour of pigs is conflicts between individuals resulting in physical injuries, conventionally divided by severity into medium and severe injuries (Table 1) [19]. Certain authors have posited that there are more than 15 variants of aggressive behaviour [20].

## 3. Physiological Markers Associated with the Manifestation of Aggression

The evaluation of physiological markers in hormones, cells, and blood associated with aggressive behaviour in pigs can serve as a valuable diagnostic tool for the early detection of potential aggression. Hormonal fluctuations, particularly cortisol levels, have been shown to modulate the tendency to aggression [22]. Cortisol is a hormone that is produced in the adrenal cortex [19]. It is responsible for important physical reactions, such as stress, and for maintaining homeostasis [23]. In pigs that exhibit aggressive behaviour, there is an increase in cortisol levels and blood pH before and during transport to the abattoir [24].

The presence of specific immune cells or inflammatory markers in the blood can provide information on the physiological state of pigs exhibiting aggressive behaviour. Serotonin represents one of the potential markers of stress [25]. A correlation has been observed between reduced serotonin levels in blood platelets and aggressive behaviour in pigs [26]. Research has shown that piglets with lower levels of serotonin in the blood are more likely to inflict damage on the tails of other piglets in the group. The aforementioned study demonstrated that piglets, fed diets with elevated tryptophan levels (the primary precursor of serotonin), exhibited reduced levels of aggression [26]. Additionally, dietary adjustments during gestation, such as providing high-tryptophan diets to sows, may positively affect the behaviour and welfare of their offspring [27]. This finding suggests that the regulation of serotonin levels, including through dietary interventions, may represent a viable strategy for the treatment of aggression.

Morphological changes that could be considered markers of aggression in pigs should be subjected to more careful scrutiny. To date, no clear morphological changes indicative of aggression in the animal have been reported. However, it has been hypothesized that alterations in lymphocyte activity and morphology may serve as potential markers of aggression. Research has shown that alterations in the dimensions and the number of nucleus organizers within lymphocytes can serve as indicators of the immune response in women, both prior to and following a pregnancy. These alterations may be concomitant with stress and manifestations of aggression [28].

The evaluation of biochemical markers, including but not limited to glucose and lactate levels, has the potential to improve our understanding of the metabolic status of aggressive pigs. Elevated levels of these markers may indicate increased energy expenditure associated with frequent aggressive behaviour [24,29].

### Environmental Factors (Addition of Novel Animal Groups, Limited Farm Space)

Changes in environmental conditions are comparable to other factors that cause aggression. These changes can be defined as alterations in the internal, genetic, and epigenetic characteristics of a single organism.

The primary focus of aggressive behaviour is the regrouping process and the stress associated with transporting animals. The manifestation of aggressive behaviour is observed to occur as a novel hierarchical structure, which is formed within the expanded group. This behaviour may be triggered by external stimuli, such as noise [30]. Noise constitutes another stressor, exerting a detrimental effect on pig behaviour and health of pigs. Noise-induced stress has been shown to result in neuronal apoptosis and general inflammation in the amygdala [31]. In contrast, music has been shown to reduce noise-induced anxiety to some extent [31]. The use of music in agricultural domains represents an unconventional, yet potentially viable, solution to the prevailing issue [32,33,34]. Music has been shown to have a positive effect on the welfare of livestock when used as an enriching factor in its environment [33]. A wide range of emotional responses has been observed in pigs due to the musical structure employed, suggesting that music may have a role to play in reducing aggression during the adaptation period [34].

Another factor that induces stress and aggression later in life is nutrition [35]. The addition of multicarbohydrate and phytase complexes to the diet provides improved nutrient digestibility, thereby promoting optimal growth performance and bone mineralisation during the critical phase of piglet development. This, in turn, can positively influence behaviour in later life, contributing to the well-being of the mixed group [36]. Research has shown that supplementation with micronutrients and substances, including magnesium, tryptophan (a precursor of serotonin), vitamin E, aromatic plant extracts, chitosan, and L-glutamine, in food has been shown to effectively reduce levels of aggression, stress, and anxiety in livestock [37,38,39,40,41]. Even antibiotic administration (specifically, chlortetracycline and tiamulin) has been shown to engender a favourable effect on the reduction in aggression, as evidenced by a decrease in the incidence of bites [37]. It has been hypothesized that discrepancies in animal behaviour may be attributable to fluctuations in the composition and diversity of the gut microbiome, which are influenced by dietary patterns, even over a brief period [37]. On the contrary, the incorporation of anabolic steroids into animal feed has been shown to be associated with the accumulation of these substances in various organs, in addition to an increase in aggressive behaviour [42].

The heat load has been similarly identified as a significant stress-inducing factor. Following exposure to heat stress ranging from 33.6 ± 1.8 °C to 37.4 ± 2.1 °C, there was a decrease in food intake and an increase in inflammatory biomarkers, suggesting the occurrence of inflammatory processes within the gastrointestinal tract [43].

The housing area also affects the social behaviour of pigs [44]. An increased housing area has been shown to reduce stress levels and the likelihood of aggressive interactions between animals. The optimal housing area for pigs, depending on the weight of the pig, is 0.24, 0.44, 0.64, 0.78, and 0.80 m^2^/per pig for live weights of 11 to 25, 25 to 45, 45 to 65, 65 to 85, and 85 to 115 kg, respectively [45]. The combined effect of animal density and temperature is also observed. The most unfavourable conditions are considered those in which high density (less than 0.50 m^2^/100 kg) is combined with high temperatures (more than 24 °C) and low density (more than 0.83 m^2^/100 kg) is accompanied by low temperatures (less than 10 °C) [46].

A significant factor in the development of aggressive behaviour is the presence of compromised immunity in animals, which can result in the onset of various health problems, including respiratory, enteric, and locomotory diseases [47]. It has been hypothesized that two distinct mechanisms may underlie the emergence of aggressive behaviour (more includes tail lesions). Firstly, the presence of a disease has been shown to result in a reduction in immunity and a decrease in the absorption of micronutrients from food, which has been shown to provoke an increase in aggression. Alternatively, pathogens can spread systemically through the body via damaged tails, reaching the lungs and lymphatic system and leading to osteomyelitis, arthritis, and lung damage, among others [47].

## 4. Genetic Predisposition and Genetic Markers of Aggression

The manifestation of aggressive behaviour in pigs, particularly in mixed groups, has been shown to be directly related to a variety of genetic factors. To date, research has indicated that aggression is an inherited behavioural trait [48,49]. For instance, the rates of inheritance for nonreciprocal and reciprocal aggression in pigs are 0.17 and 0.46, respectively. Furthermore, the rate of maternal aggression (infanticide) is 0.4 to 0.9 [50,51]. In turn, there is a demonstrable variance in the degree of aggression exhibited by different breeds. This finding suggests the presence of a genetic factor that modulates aggressive behaviour [52].

Genetic factors (markers) of aggressive behaviour play a significant role in the process of mixing and breeding of pigs to produce new offspring. A comprehensive understanding of the genetic underpinnings of aggression is instrumental in the formulation of novel breeding programmes aimed at mitigating aggressive behaviour. The implementation of improved breeding programmes can result in alterations to group behaviour, thereby enhancing animal welfare and mitigating stress levels. It should be noted that stress levels have a direct impact on the quality of meat production [24,53]. It is important to note that, in addition to aggressive behaviour, one of the significant economic challenges in pig production is the prevalence of porcine stress syndrome, whose etiology is well understood and is attributed to mutations in the ryanodine receptor gene (*RYR1*, *Arg614Cys*) [54]. The protein product has been determined to regulate calcium transport into muscle cells, and the prevalence of a pathogenic allelic variant in the *RYR1* gene has been found to be approximately 6.6% (based on data from an Argentine population) [55]. Another disease in pigs that has been linked to increased aggression is cryptorchidism. The genetic cause of this condition includes nucleotide changes in 63 genes belonging to molecular pathways, including those involved in estrogen signalling, cytoskeletal organization, and the pentose phosphate pathway [56]. Consequently, genetic screening for genetic variants and mutations associated with the development of porcine stress syndrome and cryptorchidism may represent a widespread step in the diagnosis of disorders associated with the development of aggression in pigs.

### 4.1. Candidate Gene Studies of Aggression in Pigs

Analogous to the study of genetic markers associated with interindividual variations in the level of aggression in humans, one method of identifying such loci is to analyze the associations of single-nucleotide polymorphisms (SNPs) located in candidate genes. Neurotransmitter systems are among the most extensively studied molecular systems that determine the formation of behavioural differences between humans and other mammals, including pigs. Specifically, the dopamine receptor gene (*DRD2*) has been shown to be implicated in aggressive behaviour in pigs [57]. An analysis of promoter activity has revealed that the major promoter region of the *DRD2* gene is located between positions −2212 and −1127 from the transcription initiation site. Two single-nucleotide polymorphisms (*rs1107428594* and *rs1110730503*) were identified in the promoter region of the gene as being associated with aggressive behaviour in weanling piglets. It was also revealed that *IRF1* and *IRF2* (*IRF1* regulates gene expression by binding to interferon in its promoters, while *IRF2* partially suppresses its activity) were crucial regulatory factors for *DRD2* transcription. The rs11101030503 located at the *IRF2* binding site influenced *IRF2* binding to the *DRD2* gene promoter, regulating the level of mRNA expression and protein level that could be the cause of changes in aggressive behaviour in pigs. Moreover, the results of the aforementioned study are corroborated by the established association between *DRD2* and aggressive behaviour in chickens [58].

Genes belonging to the serotonergic system are among the most extensively studied in detail with respect to their association with behavioural traits in humans and other vertebrates [59,60,61]. Furthermore, an analysis was conducted on the role of the serotonin transporter gene (*SLC6A4*) in the development of aggressive behaviour in pigs [62]. The researchers did not restrict their investigation to the study of the insertion/deletion polymorphism, which is the most studied variant in the human population and is located in the promoter region of the gene but rather extended their analysis to encompass eight genetic markers dispersed across various regions of the gene, including the promoter region, coding region, and 3′-UTR. Confirmation has been provided regarding the effect of six genetic loci; among these, rs332335871 (c.*1586G>A, located in the 3′-UTR) and rs345058216 (c.-1694C>G, located in the 5′-UTR) were identified to have particular significance [63,64]. The first SNP determines the differences in the binding of miR-671-5p microRNA, which inhibits the expression of the SLC6A4 gene in the presence of rs332335871 G-allele, while no miR-671-5p binding with SLC6A4 mRNA was observed in the case of A-allele [62]. It should be noted that miR-671-5p was previously reported to be highly expressed in excitatory neurons and involved in sensorimotor gating and synaptic transmission, which can explain its significant role in developing various psycho-emotional states and diseases [65]. In turn, the rs345058216 G-allele determines the highest binding efficiency of the transcription factor MAZ, which results in the upregulation of transcription levels of the porcine SLC6A4 gene [62]. In addition, the authors reported a link between rs345058216 GG homozygotes and higher aggressive behaviour in pigs (longer duration of fight and number of attacks) compared to CC genotype carriers. From another side, rs332335871 GG homozygotes demonstrated less duration of aggression compared with AA homozygotes. Both observations are congruent with existing findings on a link between a depletion of the SLC6A4 gene or its diminished expression and reduced aggression [66] accompanied by exaggerated levels of serotonin in the synaptic cleft.

The association of other neurotransmitter system genes with behavioural differences between two breeds of pigs (Chinese indigenous Mi pigs and Landrace–Large White cross pigs) was also examined. The first is characterized by reduced aggression (measured by backtest and skin lesion scores). The authors successfully identified specific haplotypes in the *DBH*, *HTR2A*, *GAD1*, *HTR2B*, *MAOA,* and *MAOB* genes. These were found to occur at a significantly higher frequency in more aggressive pig breeds (LLW) than in the Chinese indigenous Mi breed [67]. Haplotypes in monoamine oxidase A (*MAOA*) and dopamine β-hydroxylase (*DBH*) genes were found to be most associated with an increased risk of aggression, characterizing a 12- and 23-fold increase in the odds ratio of manifesting aggression in carriers of these haplotypes, respectively.

As one of the first genes associated with the manifestation of familial forms of aggression in humans is the *MAOA* gene [68], studies on genetic markers of aggression in pigs have also focused on the analysis of the *MAOA* gene. The study on the association of piglet aggression with the allelic frequency of nine SNPs in the *MAOA* gene showed significant differences between aggressive and docile piglets [69]. Association analysis demonstrated that pigs inheriting wild-type genotypes exhibited more aggressive behaviour than pigs with the mutant genotype of four linked SNPs: rs321936011, rs331624976, rs346245147, and rs346324437. Moreover, the GCAA haplotype was found to be associated with higher levels of aggression compared to the GCGA and ATGG haplotypes. Additionally, the plasmid construct containing the wild-type GG genotype of rs321936011 exhibited reduced promoter activity compared to the mutant AA genotype. These findings suggest that the four linked SNPs in the *MAOA* gene can be used as molecular markers for the selection of behavioural traits in pigs. The findings of this study are consistent with the observations of other researchers who have shown that variations in the *MAOA* gene in humans and mice are associated with aggressive behaviour [70,71]. An intriguing observation linking the presence of *MAOA* allelic variants with lean content and intramuscular fat was made by E. Terenina et al. [72]. Furthermore, a link has been demonstrated between high growth rate, lower fat content (leaner carcass), and higher activity and aggression in pigs [73].

The hypothalamic–pituitary–adrenal (HPA) axis is a key system involved in the regulation of aggressive behaviour in mammals, including pigs. This axis is involved in the regulation of the body’s stress response, and it is hypothesized that the stress response and the manifestation of aggression are interrelated in pigs. Several studies have aimed to identify genetic markers on the HPA axis that are potentially associated with the level of aggression in pigs. An analysis of 10 polymorphic variants of HPA axis genes (*CRH*, *CRHR1*, *CRHBP*, *POMC*, *MC2R*, *NR3C1*, *AVP*, *AVPR1B*, *UCN*, and *CRHR2*) [74] demonstrated the association between the c.*2122A>G in the glucocorticoid receptor gene (*NR3C1*) and increased cortisol concentration, increased adrenal gland mass, and a higher number of body injuries in animals. Analogous associations for SNPs were demonstrated in the arginine-vasopressin 1B receptor gene (*AVPR1B*, c.1084A>G) and in the urocortin gene (*UCN*, g.1329T>C). The results obtained are to some extent consistent with the previously demonstrated effect of the *AVPR1B* and *NR3C1* genes on the manifestation of aggression in humans [75,76]. The function of the HPA axis is also associated with a mutation in the *RYR1* gene (c.1843C>T), which determines the development of porcine stress syndrome [77].

A recent study using a candidate gene approach investigated the *JARID2* gene, which encodes the Jumonji and AT-rich interaction domain-containing protein 2 and is involved in the differentiation of progenitor cells into neurons [78]. The gene plays an important role in embryonic development in mice [79], while genetic variants in this gene are associated with the manifestation of autism spectrum disorder in humans [80], which is characterized by an increased aggression. The results obtained by comparing the frequencies of the rs3262221458 genotype in the *JARID2* gene between the most aggressive and least aggressive pigs show that the TT genotype is a risk marker for aggressive behaviour [78]. The selection of the SNP studied in the *JARID2* gene was deliberate, due to its location in the 3′-untranslated region of the gene, which plays an important regulatory role in its expression. Furthermore, genetic engineering manipulations carried out with different alleles of rs3262221458 confirmed its importance in regulating gene expression: the T allele was associated with increased gene transcription and weaker binding with the microRNA miR-9828-3p. The latter binds to rs3262221458, thus inhibiting the transcription of the *JARID2* gene [78]. This finding indicates that the rs3262221458 locus may serve as a genetic marker for aggression in pigs, thus underscoring its functional role in regulating gene expression.

Another promising candidate gene for studying aggressive behaviour in pigs is the *ARHGAP24* gene. This gene, which encodes the Rho GTPase-activating protein 24, plays a crucial role in the regulation of axonal guidance [81] and the development of the nervous system [82]. A potential role for *ARHGAP24* gene is attributable to the impact of axonal guidance on developing aggressive behaviour in humans [83] and the growth rate in pigs [84]. An association was identified between nine SNPs at the single-loci level and at the haplotype level with aggressive behaviour in pigs [85]. The rs335052970, which is located in the promoter region of the *ARHGAP24* gene and associated with aggressive behaviour, is of particular importance. The rs335052970 A allele results in an increased binding to the p53 transcription factor, resulting in a diminished transcription of the *ARHGAP24* gene itself. Consequently, rs335052970 can be used as a genetic marker of aggression [85].

### 4.2. Genome-Wide Linkage Studies of Aggression in Pigs

In addition to the identification of genetic markers associated with the manifestation of aggressive behaviour in pigs that have been mixed for 24 h in a pen, it is of relevance to determine genomic regions and markers associated with such a type of aggressive behaviour in sows as infanticide. Using the genome-wide linkage analysis, which evaluated a linkage of 80 microsatellite markers located on 18 autosomes and the X chromosome with the propensity for infanticide, four QTL loci located on chromosomes 2 (SSC2), 10 (SSC10), and X (SSCX) were identified [86]. The determined chromosomal regions in pigs are syntenic with the chromosomal regions 5q14.3-15, 1q32, Xpter-Xp2.1, and Xq2.4-Xqter in humans, in which the localized genes have previously been associated with human behaviour and emotional states [87,88,89,90,91]. The results obtained in pigs indicate the potential contribution of such genes as *STS* (located in the short arm region of the X chromosome), *PGRMC1* (located in the long arm region of the X chromosome), *PTPRC* (chromosome 10 region), and *COX7C* (chromosome 2 region) to the manifestation of aggressive behaviour in sows. The identification of these chromosomal regions is not accidental, as other studies have also indicated the potential contribution of the aforementioned genes to the regulation of behaviour. In particular, the level of steroid sulfatase, encoded by the *STS* gene and involved in the regulation of neurosteroid levels, was associated with the manifestation of aggression in rodents [92]. The homologue of the progesterone receptor membrane component 1 (encoded by the *PGRMC1* gene) is structurally similar to the prolactin receptor and is coexpressed with arginine vasopressin in the hypothalamus [93], which may indicate its indirect effect on the regulation of aggressive behaviour. Moreover, the *PTPRC* gene encodes the tyrosine phosphatase, receptor type C, a signalling molecule necessary for the regulation of cytokine signalling and neuroinflammation, the impairments of which are also related to behavioural fluctuations [47,94]. One of the potential candidates located in the region on chromosome 2 is the *COX7C* gene, which encodes the cytochrome c subunit VII. It is acknowledged that cytochrome c plays a regulatory role in the mitochondrial respiratory chain and that disturbances in this chain have been implicated in the development of mental disorders [95].

### 4.3. Genome-Wide Association Studies of Aggression in Pigs

Another method that enables the detection of genetic variants associated with aggressive behaviour (AB) is the genome-wide association study (GWAS), which simultaneously analyzes tens to hundreds of thousands of SNPs for their association with a phenotype. This method has become especially popular because it does not require knowledge on specific molecular systems potentially involved in the development of the trait studied. To date, the number of GWASs that address aggressive behaviour in pigs remains scarce. However, these studies have identified four SNPs located on chromosome 11 and are associated with the infliction of damage to the front and central parts of the body of Yorkshire pigs [96]. The most significant genetic markers (*ALGA0061562* and *M1GA0026237*) explained up to 6.47% of the phenotypic variance in the level of aggression. The neighbouring genes are *VWA8* (von Willebrand factor a domain containing 8), *DGKH* (diacylglycerol kinase eta) and the unknown protein-coding gene *ENSSSCG00000009430* [89]. Existing results confirm studies in humans and rodents to some extent, indicating the involvement of the *DGKH* gene in variances in behaviour [97].

The capabilities of a GWAS are mainly limited by verifying biallelic genetic variants, including single-nucleotide insertions/deletions. Currently, copy number variations (CNVs) of genomic regions can also determine the development of multifactorial traits [98]. A genome-wide screening of CNVs in pigs has identified 6869 variants, including an increase in CNVs in the *SLCO3A1* gene, which has been linked to exaggerated aggression and composite aggressive score in pigs [99]. The Organic Anion Transporter Family of Solute Carrier 3A1 (*SLCO3A1*) belongs to a family of transporters that facilitate the penetration of substances into cells. Additionally, it is assumed that this protein in signal transduction processes between neurons [100] and neurotransmission is mediated by the neuropeptide vasopressin [101]. Furthermore, a differential expression of the *SLCO3A1* gene was observed in the temporal lobe between high and low-aggressive animals, which potentially indicates a link between CNVs in the *SLCO3A1* gene and its transcription. In particular, individuals with aggressive behaviour exhibited a significantly higher frequency of loss-type CNV variants compared to gain-type CNV variants, which has been corroborated by several studies [99,102].

### 4.4. Gene Transcription as a Marker of Aggressive Behaviour in Pigs

Alongside the identification of specific genetic markers of aggressive behaviour in pigs, it is also important to determine the associated transcriptional profile that characterizes gene expression. It is acknowledged that the brain volume of the domestic pig is reduced by approximately 18% compared to that of the wild boar as a result of domestication [103]. Consequently, it can be hypothesized that there are differential changes in the level of gene transcription caused by this evolutionary process, including reduced animal aggression.

Ideally, the expression level is carried out in the brain tissues (in particular, in the prefrontal cortex), allowing a direct estimation of gene activity in the brain regions responsible for behavioural regulation. One such study was based on an assessment of the expression level of genes associated with the domestication of the wild boar, a decrease in its level of aggression, and the formation of the domestic pig [104]. The comparative analysis of transcriptomes of the wild boar and domestic pig carried out within this study revealed 60 differentially expressed genes related to the immune response system, while 7 genes were associated with the domestication of this animal species. A comparison of the panel of differentially expressed genes with the previous study [105], which aimed to identify the link between domestication and the level of transcription in various species of domestic and wild animals, including pigs, dogs, guinea pigs, and rabbits, resulted in similar expression profiles of the annexin (*ANXA*) and carboxypeptidase (*CPXM*) gene families in domesticated pigs. Furthermore, a comparison of the transcription profile of pigs with guinea pigs and rabbits revealed an overlap of five genes (*C7*, *ACSM5*, *DSC2*, *C7orf61*, and *CHRNA6*) [104], suggesting their potential involvement in domestication linked to reduced animal aggression. It should be noted that the *C7* gene encodes a component of the complement system, which plays a crucial role in the regulation of innate immunity and the immune system [106]. This system is actively studied in the context of its role in regulating human behaviour, including aggression and similar phenotypes [107,108]. A further large-scale study, which systematized the results of 223 epigenome and transcriptome data obtained for 14 different pig tissues, enabled the identification of differentially expressed genomic regions that characterize the emergence of domestication processes in different breeds of animals. Specifically, in European breeds that exhibit higher levels of aggression [67], such differential changes associated with domestication affected genomic regulatory elements in the tissues of the cerebral cortex. On the contrary, the differential expression of regulatory elements was observed in Asian pig breeds in the spleen [109]. Another significant indicator of the regulatory changes that accompany the wild boar is the level of expression of regulatory circular RNAs (circRNA). Analysis of the expression of more than 11,000 different circRNAs in the prefrontal cortex of animals allowed the identification of the circRNA ssc_circ_6179, which was characterized by a differential level of expression between the wild boar and the domestic pig [110]. Furthermore, ssc_circ_6179 has been shown to play a regulatory role in synaptic activity via its binding to microRNA miR-9847, which in turn suppressed *HRH3* expression.

In addition to the involvement of neurotransmitter system genes in the development of aggressive behaviour in pigs at the genetic marker level [57,67,69], there is evidence to their transcriptional changes. Specifically, the administration of the β-adrenoreceptor agonist ractopamine (RAC) to pigs resulted in aggressive behaviour, which was associated with an altered transcription of the dopamine D1 receptor in brain regions, including the raphe nuclei and frontal cortex [111]. Additionally, the authors demonstrated sex-specific differences in the transcription levels of serotonergic system genes in the swine brain, including the *HTR1A*, *HTR2A*, *HTR2B*, and *MAOA* genes. These factors, among others, may account for differences in aggression levels between animals (Figure 1).

In addition to assessing the transcriptional changes that are characteristic of different regions of the brain, several studies have focused on analyzing the transcriptome in peripheral tissues, including the adrenal glands. A study detected an association between psychosocial stress, induced by mixing groups of unfamiliar individuals, and the high expression of genes related to cholesterol accumulation and an association between the precursor of the miR-202 microRNA and the low expression of genes involved in the regulation of cell growth and apoptosis [112].

When implementing genetic selection or modification to reduce aggression in pigs, adherence to ethical standards is essential to ensure animal welfare and avoid unintended consequences. Responsible practices should align with established guidelines, such as the “Three Rs” (Replacement, Reduction, Refinement), to minimize harm and prioritize humane treatment in livestock breeding [113].

## 5. The Link Between Environmental-Induced Epigenetic Changes and Aggressive Behaviour

One of the potential mechanisms linking the manifestation of behavioural traits with certain environmental influences is epigenetic changes that cause differences in the transcriptional activity of genes [114]. Although existing findings unravelling a link between the effects of environmental stressors on genes’ activity related to certain genetic markers remain scarce, their possible impact on behavioural changes is assumed to be caused by epigenetic changes, which regulate the transcriptional activity of specific genes. Recently, it was demonstrated that environmental stressors such as the barren environment of sows and their repetitive behaviour caused significant changes in the epigenome of their piglets, which was analyzed in various brain regions including those related to emotionality, learning, memory, and stress response [115]. An enrichment analysis of the top 16 differentially methylated regions (DMRs) revealed the genes, which belonged to neural crest development, alcohol and amyloid metabolism, lipid-mediated signalling, and microtubule poly/depolymerization pathways. Interestingly, such neuro-epigenetic effects in the hippocampus and frontal cortex of piglets were mainly attributed to the scarce environment of sows, while maternal stereotypic behaviour predominantly affected the amygdala epigenome and was previously reported to be related to piglets’ emotionality [116]. So far, it can be suggested that the carriers of alternative allelic variants of a gene marker, which was shown to be linked to differential gene expression, may result in the epigenome-driven altered vulnerability of carriers of certain alleles due to the effect of environmental stressors, as was previously reported in humans [117]. However, distinct mechanisms linking environmental influences, allelic variants, gene expression, and behavioural alterations represent a challenging field of future research.

The data on the favourable/unfavourable effects of certain factors on changes in gene transcription can be used to model behaviour in pigs [118]. The existing stressful conditions in the early postnatal period remain a significant factor determining future differences in behaviour, including aggression and anxiety. In particular, early maternal weaning and social isolation caused subsequent significant changes at the transcriptome level, including genes involved in the regulation of gene methylation levels, neurogenesis, HPA axis, synaptic plasticity, resulting in the development of excessive anxiety and aggression in model animals [119,120]. An examination of the expression levels of HPA axis genes in the prefrontal cortex and hippocampus of such piglets revealed a decrease in the expression of glucocorticoid receptor genes (*NR3C1*), mineralocorticoid receptor genes, and 11beta-hydroxysteroid dehydrogenases (*11beta-HSD2*) in the hippocampus of early weaned piglets [121,122]. A similar pattern was observed in piglets subjected to social isolation: a reduced expression level of these HPA axis genes (*11β-HSD1* and *2*, *NR3C1*, *NR3C2*) was observed in the prefrontal cortex regardless of age [121]. Subsequently, a diminished expression of the *NR3C1* gene results in the attenuation of negative feedback causing an inhibited activation of the HPA axis and glucocorticoid synthesis [123]. Disrupted HPA axis negative feedback is linked to lower expressions of the *NR3C1* gene in the hippocampus and impaired brain functioning [123]. Such down-regulation of the *NR3C1* gene expression is attributed to post-weaning hypermethylation of the promoter region [124]. Therefore, early-weaned piglets can be affected by raised levels of glucocorticoids, which may result in adverse effects on cognitive traits, self-control, and self-regulation of behaviour [125]. Moreover, social isolation also causes alterations at behavioural, neuroendocrine, and immune levels due to modified HPA axis activity (for example, glucocorticoid binding) [126] and changes in neurotransmitter functioning [127].

Additionally, maternal immune activation during pregnancy may have a detrimental effect on fetal development in the postnatal period through molecular changes in the amygdala. In particular, when comparing two groups of piglets, one fed by a dam with increased immune activation due to viral load and the other fed by a non-native sow without viral infection, significant changes in gene expression related to inflammation, immune response, glutamatergic regulation, and neurological disorders according to KEGG were found in the amygdala of 3-week-old piglets [118,128]. Such changes in glucocorticoid levels and the inflammatory mediation system may subsequently affect the development of negative behavioural traits [129].

## 6. Problems of Strategy Implementation

Despite the relatively high level of understanding of the genetic and other factors that cause aggression, the strategy to minimize such behaviour remains a significant challenge in modern agriculture.

The identification of aggression liability, which represents a polygenic trait, cannot be based on the detection of a single specific marker. The simultaneous influence of multiple genes can affect behaviour in different ways, making this field a promising area for further research. In this regard, a novel approach is proposed, namely, the use of a multitrait selection index. This index is based on the clustering of various behavioural traits reflecting aggression levels, thereby facilitating further actions to its decline [130].

A persistent challenge that persists even after the implementation of novel genetic strategies in agricultural practice is related to the reluctance of farmers to adopt new technologies. Farming enterprises may be hesitant to allocate resources towards the integration of “genetic” management methodologies, despite their potential to confer benefits such as enhanced animal welfare and reduced aggression-related trauma, if the financial incentives are deemed inadequate. Farmers may opt for more conservative practices if the anticipated benefits do not outweigh the costs, thereby reducing the financial incentive for them to implement these strategies [131].

Notwithstanding the encouraging results emanating from the use of genetic markers of aggressive behaviour in swine, this methodology is concomitant with challenges related to genetic intricacy, shifts in behavioural dynamics, the conservatism of management techniques, and economic factors. To address these challenges, a holistic approach is imperative, entailing the integration of genetics with effective management methodologies, while concomitantly taking into account the perspectives of farmers with regard to the associated costs and benefits.

### Future Prospects

The field of genetic selection and the use of genetic markers for aggressive behaviour shows significant developmental potential due to several key areas of research. The use of molecular genetic markers for aggression opens new horizons in selection, allowing for a more precise selection of individuals for further breeding. Increased attention to genetic markers and subsequent development in this field will lead to improvements in the physiological condition of animals, enhanced product quality, and increased market demand for such meat. However, assuming existing findings, the use of genetic data to select for liability to aggressive behaviour can be made with caution. The first point is that it is impossible to emphasize the major genes/genetic systems, which have the highest contribution to manifest increased aggression in pigs. Serotonergic system genes (like *SLC6A4*, *MAOA*) are the most extensively studied ones with respect to aggression in different species, including pigs. Although a plethora of studies aimed to identify specific genetic markers of swine aggression implementing various molecular approaches has been carried out, summarized results remain ambiguous. To make more valuable conclusions on the practical use of genetic data, it is of relevance to conduct replication studies of various genetic markers accounting for breeds’ specificity (i.e., European and Asian pig breeds) of behaviour. Analogous to the polygenic score (PGS) approach, which summarizes the effect sizes of different genetic variants based on genetic profile, widely used to predict an individual’s liability to complex traits in humans, implementing the PGS based on data from multiple SNPs can elucidate certain liability to manifest excessive aggression in pigs. The use of the PGS to predict swine aggression is challenging; however, it can help with elaborating genetic screening to exclude predisposed subjects from breeding programmes.

The genetic selection of pigs on the basis of alternative traits is of considerable importance, in particular, in evaluating the presence of “risk” allelic variants of genetic markers associated with high aggression. In particular, pigs selected for higher growth rates showed 27–40% less aggression toward various parts of the body [132]. In addition to the use of selection to fix desired phenotypic traits in pigs, it is of particular interest to obtain transgenic animals for xenotransplantation. However, to date there are insufficient studies to assess the impact of introduced transgenic constructs on behavioural changes in transgenic animals, including aggression. However, results from one study suggest that transgenic animals carrying the human decay-accelerating factor gene do not differ from conventional pigs in terms of aggression levels, social interactions, cortisol levels, and other parameters [133], indicating the potential for breeding transgenic pigs without signs of aggression.

The development of genetic modification technologies, accelerated by the availability of these markers, promises to promote the creation of new breeds with optimal characteristics. This, in turn, may cause the emergence of new pig breeds that are more adaptable to new conditions, thereby ensuring the long-term sustainability of the industry. The reduction in pig aggression is expected to result in a significant decrease in injuries to both animals and farm workers, consequently reducing veterinary costs and improving the overall economic efficiency of production.

Emerging computational methods are expanding our capacity to link genetic markers with complex social behaviours. Among these, the social network analysis offers a transformative lens for understanding aggression’s genetic basis by quantifying how individual behaviour shapes group dynamics. For example, identifying main aggressors—pigs whose *DRD2* or *MAOA* genotypes correlate with disproportionate social influence—could refine breeding selection. However, farm-scale implementation requires cost-effective tracking technologies and standardized metrics to translate network patterns into action-able genetic insights.

While beyond the scope of this review, strategic crossbreeding between high- and low-aggression breeds warrants investigation as a complementary approach. Such efforts should prioritize longitudinal studies of hybrid behaviour and economic viability assessments in commercial settings.

## 7. Conclusions

The pig genetic breeding industry faces some challenges when it comes to reducing aggression in pigs. There are genetic (including specific markers like *DRD2*, *SLC6A4*, and *MAOA*), environmental (covering housing conditions, nutrition, and stress factors), and economic factors (implementation challenges, farmer adoption, and barriers to adopting genetic screening) that make it difficult to implement effective strategies. However, even though there are challenges, the future of the industry looks bright because of molecular markers and new technologies. These tools can help improve the welfare and behaviour of pigs, as well as make pig production more efficient. This will make the industry more resistant to changes in market demands. 

## Figures and Tables

**Figure 1 genes-16-00534-f001:**
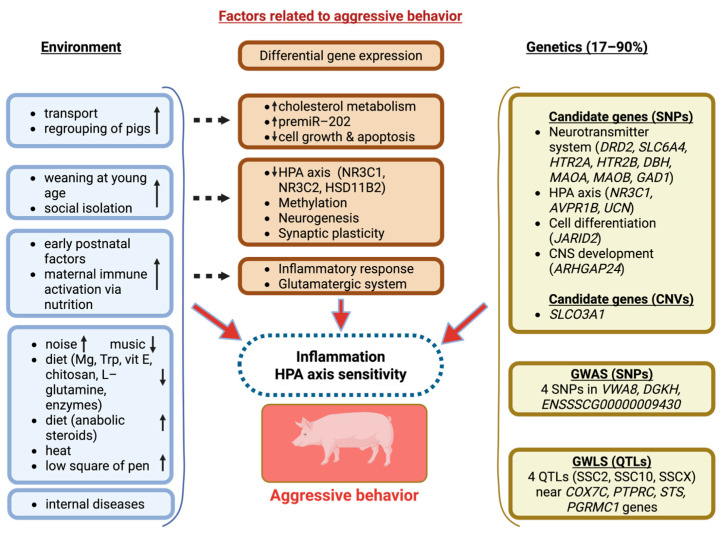
Environmental and genetic factors that contribute to aggressive behaviour in pigs.

**Table 1 genes-16-00534-t001:** Classification of the severity of damage caused by different manifestations of aggressive behaviour.

Severity of the Damage Caused	Variant of Aggression	Description
Medium	Head-to-head impact	The act of striking the cranium, muzzle, or neck of another animal without employing the mouth is referred to as “striking the head/muzzle”.
	Head-to-body impact	Impact of a blow to the area posterior to the ears of another animal.
	Parallel pressing	Animals exhibit a lateral-to-lateral comparison of their bodies, characterized by the application of pressure through their shoulder girdle regions, accompanied by their craniums being positioned posterior to one another’s necks or crania.
	Reverse parallel pressing	The animals adopt a position opposite one another followed by a force with their shoulder girdle, causing their heads to move behind each other’s neck or head.
Heavy	Neck bite	The neck bite was characterized by its aggressive nature, without a concomitant head thrust.
	Body bite	Aggressive bite to the body.
	Ear bite	Aggressive biting of another pig’s neck.
	Tail bite/tail bite off	The range of interactions with the tail encompasses a variety of severity levels, from minor nicks to severe damage to the sacrum [21].

## Data Availability

Not applicable.

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
