# Peer review of "Genetic Contributions to Aggressive Behaviour in Pigs: A Comprehensive Review"

_genes, 2025, doi:10.3390/genes16050534_

Round 1

Reviewer 1 Report

Comments and Suggestions for Authors

The authors present a comprehensive review on aggressive behaviour and its genetic components. 

An issue not addressed are obvious breed differences in the aggressive behaviour. The review should consider in which breeds the respective studies were performed. In commercial pig breeds aggression my be much more a problem in comparison to traditional, often autochthonous pig breeds. There may be large differences.

Crossbreeding would be an other possibility to reduce aggression.

Breeding for good behaviour: what are the main goals, e.g., reducing tail biting and aggressive behaviuor.

Image analysis may help to record aggressive and damaging behaviour.

Also, improved statistical models like social network analysis may be a further mosaic stone.

It may be questioned if genetic markers can solve the breeding problem. Genomic breeding values have to be discussed as well as the models which should consider social interactions.  

Author Response

Responses to Reviewers’ comments on the manuscript by Kazantseva et al. entitled “Genetic contributions to aggressive behavior in pigs: a comprehensive review”

We would like to thank the Reviewers for their thorough review of our manuscript. To address the reviewers’ concerns and excellent suggestions, we have revised the text extensively based on the Reviewers’ comments. We think that these changes have substantially strengthened the manuscript. We respond in detail to all the Reviewers’ concerns below.

Reviewer’s comments to the Author:

General comment

The authors present a comprehensive review on aggressive behaviour and its genetic components.

Response

We thank the Reviewer for finding our approach innovative and the conclusions from its application potentially acceptable.

Comment 1
An issue not addressed are obvious breed differences in the aggressive behaviour. The review should consider in which breeds the respective studies were performed. In commercial pig breeds aggression my be much more a problem in comparison to traditional, often autochthonous pig breeds. There may be large differences.

Response

We sincerely appreciate the reviewer's insightful comment regarding breed-specific differences in aggressive behavior. We agree this is a critical consideration for understanding the genetic basis of porcine aggression and are pleased to highlight how our manuscript addresses this important aspect.

 In the section examining genetic contributions to aggression (lines 248-251), we highlight the substantial variation in aggressive behavior manifestations between different pig breeds. These behavioral differences likely reflect underlying genetic distinctions that have emerged through diverse selection pressures, whether for production traits in commercial lines or environmental adaptation in traditional breeds.

A particularly compelling case study emerges from the comparative analysis of Landrace-Large White (LLW) crosses and Chinese indigenous Mi pigs (lines 316-328). The Mi breed demonstrates significantly attenuated aggressive behaviors, a phenotypic distinction that has been molecularly correlated with specific haplotype variants in several neurotransmitter-related genes (DBH, HTR2A, GAD1, HTR2B, MAOA, and MAOB). These genetic markers occur with substantially higher frequency in more aggressive porcine populations. Notably, inheritable haplotypes in both monoamine oxidase A (MAOA) and dopamine β-hydroxylase (DBH) loci were associated with dramatically elevated aggression risk, exhibiting 12-fold and 23-fold increases in odds ratios for aggressive manifestations respectively.

Comment 2

Crossbreeding would be an other possibility to reduce aggression.

Response

The reviewer raises an important point regarding crossbreeding as a potential strategy to reduce aggression in pigs by combining genetic material from aggressive and non-aggressive individuals. While this approach holds theoretical promise, we wish to clarify that our current review focuses specifically on within-breed genetic markers and their associations with aggressive behavior, as outlined in Sections 4.1–4.4. 

Notably, the studies we analyzed did not explicitly examine crossbreeding effects. However, we emphasize that implementing crossbreeding programs would require: 

  • Systematic validation of trait stability in hybrids, as epistatic interactions may unpredictably affect behavior.
  • Cost-benefit analyses with industry stakeholders, given potential trade-offs with productivity traits (growth rate).
  • Ethical considerations, as aggressive behavior may resurface in subsequent generations without careful selection.

We agree this topic merits dedicated research and have added the following to Section 6.1 (Lines 700-703): "While beyond the scope of this review, strategic crossbreeding between high- and low-aggression breeds warrants investigation as a complementary approach. Such efforts should prioritize longitudinal studies of hybrid behavior and economic viability assessments in commercial settings."

Comment 3

Breeding for good behaviour: what are the main goals, e.g., reducing tail biting and aggressive behaviuor.

Response

We sincerely appreciate the reviewer’s important question regarding breeding objectives for improved behavior. Based on our introduction section in Lines 41–49, we emphasize that reducing aggression is not only a welfare concern but also critically tied to production efficiency and economic outcomes. In particulary elevated aggression leads to chronic stress, which reduces feed efficiency, growth rates, and meat quality. Also, economic losses stem from increased veterinary costs, higher mortality, and carcass downgrades due to injuries.

Complete elimination of aggression is neither feasible nor desirable, as some forms (play aggression) are developmentally normal. Instead, breeding should focus on reducing pathological aggressio linked to economic losses, while ensuring pigs retain adaptive social skills

The proposed breeding objectives are oriented towards achieving quantifiable reductions in high-cost aggression (e.g. tail biting, severe fights), whilst acknowledging the necessity for complementary management strategies. The propensity for aggression in pigs is underpinned by a robust genetic component. It has been demonstrated that certain genetic markers contribute to an enhanced understanding of the aetiology of deleterious forms of aggression, including, but not limited to, excessive fighting and tail biting, in swine. The integration of genetic testing into breeding programmes can allow farmers to select animals with calmer temperaments, thereby reducing aggression in subsequent generations.

Comment 4

Image analysis may help to record aggressive and damaging behaviour.

Response

We appreciate the reviewer's valuable emphasis on automated behavior monitoring systems. Our original discussion (Lines 124-133) established that while both traditional observation and emerging automated systems (utilizing machine learning and computer vision) represent important methodological approaches for recognizing and responding to aggressive behavior. To address the reviewer's concerns more comprehensively, we have augmented our discussion with new text (beginning at Line 134) that: Acknowledges the demonstrated efficacy of automated recognition systems in controlled settings; Systematically examines implementation challenges including:

  • Technical constraints in commercial farm environments
  • Economic feasibility considerations
  • Validation requirements for diverse housing conditions

Added text to section 2 (Lines 134-139): “Automated recognition systems demonstrate effectiveness in detecting aggressive behavior, despite limited interpretability of their artificial intelligence models [15]. How-ever, these systems may face significant limitations: their performance in dense herds and dynamic lighting conditions remains unverified, and the substantial infrastructure in-vestments required for high-performance systems may prove economically prohibitive for medium-sized farms.

This expanded analysis reinforces our conclusion that standardized observation protocols remain essential for reliable behavioral assessment, particularly for genetic studies requiring high-fidelity phenotyping. We emphasize this position not as a rejection of technological solutions, but rather as recognition of the need for further development and validation before these systems can fully replace established methodologies in research and breeding programs”.

Comment 5
Also, improved statistical models like social network analysis may be a further mosaic stone.

Response

We thank the reviewer for their valuable suggestion regarding social network analysis (SNA) as a methodological advancement. While our original manuscript did not explicitly address SNA, we fully agree that this approach could provide critical insights into the social dynamics of aggression. In response to this comment, we have added the following discussion to Section 6.1 (Lines 692-699): “Emerging computational methods are expanding our capacity to link genetic markers with complex social behaviors. Among these, social network analysis offers a transformative lens for understanding aggression’s genetic basis by quantifying how in-dividual behavior shapes group dynamics. For example, identifying main aggressors—pigs whose DRD2 or MAOA genotypes correlate with disproportionate social influence—could refine breeding selection. However, farm-scale implementation requires cost-effective tracking technologies and standardized metrics to translate network patterns into actionable genetic insights.”

Comment 6

It may be questioned if genetic markers can solve the breeding problem. Genomic breeding values have to be discussed as well as the models which should consider social interactions.

 Response

Thank you for raising this important point regarding the application of genetic markers in breeding programs. The section in lines 270–347 acknowledges the polygenic nature of aggressive behavior and emphasizes that while individual genetic markers (e.g., *DRD2*, *SLC6A4*, *MAOA*) provide valuable insights into biological mechanisms, their predictive power is indeed limited when considered in isolation. 

Our discussion highlights that genomic breeding values and models incorporating social interactions are already central to addressing this complexity. For example, we note how aggression manifests through both direct genetic influences and social dynamics, underscoring the need for integrated approaches. The text also references emerging methodologies like social network analysis which align with your suggestion by quantifying how genetic predispositions interact with group behavior. 

While we agree that no single solution exists, the evidence presented demonstrates that genetic markers, when contextualized within broader genomic and social frameworks, contribute meaningfully to breeding strategies.

Reviewer 2 Report

Comments and Suggestions for Authors

See the file attached.

Author Response

Responses to Reviewers’ comments on the manuscript by Kazantseva et al. entitled “Genetic contributions to aggressive behavior in pigs: a comprehensive review”

We would like to thank the Reviewers for their thorough review of our manuscript. To address the reviewers’ concerns and excellent suggestions, we have revised the text extensively based on the Reviewers’ comments. We think that these changes have substantially strengthened the manuscript. We respond in detail to all the Reviewers’ concerns below.

Reviewer (Comments to the Author):

General comment

Thank you for the opportunity given to me to review this paper. In this review, the authors took a closer look at the different factors that lead to aggression in pigs, focusing on genetic, environmental, and physiological aspects. The paper is not suitable for publication at its present state. I have suggested a few areas where it could be improved.

Response

We sincerely appreciate your time and valuable feedback on our manuscript. We are grateful for the opportunity to improve our work based on your constructive suggestions regarding the genetic, environmental and physiological aspects of pig aggression. We have carefully considered all of your comments and have made substantial revisions to address the areas you highlighted for improvement. The revised manuscript now provides more comprehensive coverage of these key factors and their interactions.

Comment 1
Abstract

  1. It touches on genetic, environmental, and physiological factors, but it doesn’t really dive into how these elements interact with each other. Understanding this interplay is key to getting a complete picture of aggression. It would be great if the abstract could in a few sentences how these factors work together to influence aggressive behaviour in different settings.
  2. Also, the section on genetic studies is interesting, but it could use more specifics about the findings and how much these genetic markers affect aggression. Including some concrete examples or results would really bolster this part.
  3. The abstract mentions advancements in genetic research but doesn’t explain how these can be applied to current breeding practices. Providing some clear examples of how to integrate these findings into existing systems would make it much more useful.

Response

Thank you for your constructive feedback. We have revised the abstract to better highlight specific genetic markers of aggression and the challenges of implementing genetic findings in practical breeding programs, while maintaining our focus on genetic markers of aggression. (Lines 25-31)

Comment 2

While citations appear in some areas, they’re noticeably absent in others. When discussing specific types of aggression or physiological factors, it would be great to see more consistent references to support those claims, especially in sections that delve into environmental and social influences.

Response

We have added relevant citations throughout the manuscript to better support our discussion of aggression types and physiological factors, particularly in sections addressing environmental and social influences. (Line 53,64,153,154,160,189,192,201)

Comment 3

The paper covers various types of aggression and their impacts, but it falls short on providing statistical data or studies to illustrate how prevalent or severe these behaviours are. Including some data would lend more credibility and make the review more persuasive.

Response

In response to your comment, we have included a paragraph that discusses the prevalence of aggression in pigs, particularly during regrouping. This section references relevant studies that provide insights into the occurrence of aggressive behaviors and their implications for animal welfare. For instance, we noted that aggression is common when unfamiliar pigs are mixed, with studies indicating that approximately 22% of victims of aggression survive due to timely intervention by farmers. (Lines 53-59).

Comment 4

While the paper is quite detailed, it can be a bit hard to follow because of the heavy amount of information packed in. Some parts, especially those dealing with genetic and molecular aspects, might feel a bit overwhelming for readers. It would be helpful to have a clearer structure or maybe a summary of the main points at the end of each section to boost understanding.

Response

Thank you for your insightful comments regarding the structure and clarity of our manuscript. We appreciate your feedback on the density of information, particularly in the sections addressing genetic and molecular aspects.

After careful consideration, we have decided to maintain the current structure of the manuscript, especially the detailed discussions related to genetics. This focus is essential, as it aligns with the title of our review and underscores the importance of genetic factors in understanding aggressive behavior in pigs. We believe that the depth of information provided is crucial for readers who are seeking a comprehensive understanding of the topic.

Comment 5
The paper does a solid job of diving into the theoretical and scientific aspects of aggression, but it could benefit from a stronger focus on practical advice or real-world applications. For instance, it touches on the role of serotonin and dietary changes, but it would be great to see more specifics on how these strategies are being used in commercial pig farming.

Response

In response to your feedback, we have added additional information to our manuscript regarding dietary adjustments during gestation. Specifically, we highlight that providing high-tryptophan diets to sows may positively affect the behavior and welfare of their offspring. This dietary strategy is supported by research indicating that increased tryptophan can lead to higher serotonin levels, which may contribute to improved animal welfare and reduced aggression. (Lines 165-167).

Comment 6

There are quite a few genetic markers mentioned, but the connections between them and aggressive behaviour in pigs could use a bit more clarity. For instance, when talking about specific SNPs like rs332335871 or rs345058216, it would be great to include a short explanation of how these markers impact aggression or give some context about their mechanisms.

Response

We have expanded reported findings on the functional significance of functional SNPs rs332335871 and rs345058216 in the SLC6A4 gene and their links with aggression in pigs (lines 299-315). The information regarding specific alleles in both SNPs and their link to diminished aggression was caused by reduced expression of the SLC6A4 gene (due to enhanced miR-671-5p binding or diminished binding of MAZ transcription factor) resulting in enhanced level of serotonin in the synaptic cleft.

Comment 7

The paper cites several studies, but it could really shine with a clearer synthesis of how these studies connect. Instead of just listing different genetic markers, it would be more effective to compare the findings directly, showing how they support or contradict each other. This would help create a more cohesive narrative on the topic.

Response

We thank the reviewer for a valuable comment. However, it seems inapplicable to compare directly the results on genetic markers of the studies conducted on different genetic systems in pigs or implementing various approaches (i.e. candidate gene studies, GWAS, GWLS, etc.), since each of authors’ group has focused on a certain set of different genetic markers. In comparison to the examination of the role of the SLC6A4 gene in developing psychoemotional traits in humans, which are mainly focused on insertion-deletion promoter polymorphism (5-HTTLPR), porcine studies implicate non-overlapping SNPs. Even genetic markers in the MAOA gene, which is the most known one linked to aggression, differ between mentioned studies. Namely, rs321936011, rs331624976, rs346245147, and rs346324437 in Chen et al. (2019) and rs81499537, rs81242206, rs81242207, rs81220383, rs80875407, rs80895596 in Alia-Klein et al., (2008). Moreover, the findings obtained via GWAS or GWLS have identified genetic markers in another set of genes compared with the genes examined via candidate gene approach. To date, a field of genetics research of pigs remains incomplete, especially, if it discovers certain genetic markers. Nevertheless, the manuscript makes a general comparison of results obtained in pigs and other species at a gene level where it is applicable.

Comment 8

Some genetic markers, like those linked to the serotonin transporter gene (SLC6A4) or the MAOA gene, are highlighted as significant, but there’s room to dive deeper into what these findings really mean. It would be beneficial to discuss how these insights could be applied, such as their potential impact on breeding programs or improving animal welfare practices.

Response

The manuscript discusses such issues in Future Prospects section (lines 650-703). Although, we have added a paragraph (lines 657-672), which describes a potential impact of genetic screening programs in pigs on their welfare. However, to date it remains impossible to select distinct genes/ genetic systems, which have a greater impact on pigs’ aggression.

Comment 9

While the focus is on genetic factors, it’s important to remember that aggression in pigs is likely shaped by a mix of genetic, environmental, and epigenetic influences. A brief mention of how these genetic markers might interact with environmental stressors or dietary factors, along with the role of epigenetics, would provide a more well-rounded perspective.

Response

To date the findings on a complex link between certain genetic markers, environmental stressors and epigenetic changes in pigs remain scarce compared to humans. Nevertheless, for better comprehension we have added existing findings to possibly clarify this issue (lines 569-589).

Comment 10

The text dives into some complex ideas, and it can come off as a bit dense, making it tough for some readers to keep up. Splitting the information into smaller, more digestible sections or subsections could really boost readability. For instance, showcasing specific studies in their own paragraphs might help clarify things.

Response

We have added subsections (4.1-4.4) for better comprehension of mentioned results.

Comment 11

The part discussing environmental-induced epigenetic changes in aggressive behaviour has some great insights, but it could use a bit more detail. For example, while it mentions early maternal weaning and social isolation as factors influencing aggression, it would be helpful to explore how these elements impact gene expression in a more thorough way

Response

We thank the reviewer for this comment and added more detailed information on a link between maternal weaning, social isolation on changes in gene expression caused by changes in HPA axis functioning (lines 603-615).

Comment 12

In the final paragraph of this section, it talks about farmers being hesitant to adopt genetic management strategies because of economic reasons. Including specific examples or case studies where these strategies have been successfully implemented would provide solid evidence to back up the claims made.

Response

Thank you for your insightful comments regarding the inclusion of specific examples of successful implementations of genetic management strategies in pig farming. We appreciate your suggestion; however, we must clarify that, as of now, there are no documented case studies that provide concrete examples of these strategies being successfully implemented in commercial settings. Despite the absence of specific examples, we believe that genetic management strategies hold significant promise for enhancing productivity and welfare in pig farming. These strategies, including selective breeding and genetic modifications, have the potential to improve traits such as growth rate, feed efficiency, and disease resistance.

Comment 13

Even though the focus is on genetic selection and behavioural traits, there’s no mention of the ethical concerns that come with genetic modification or selective breeding for aggressive traits. A quick nod to ethical considerations could really enrich the discussion, especially when it comes to animal welfare.

Response

In response to your suggestion, we have added a dedicated paragraph on ethical considerations (Lines 560-564)

Comment 14

You might notice a few minor inconsistencies in how gene names are formatted (like ARHGAP24 vs. ARHGAP24) and in the citation styles (for example, [76] and [91]). Keeping things consistent, especially with gene names and references, will really boost the professionalism and readability of the paper.

Response

The manuscript has been checked for inconsistencies in formatting gene names and citation styles. All gene names were marked as Italics.

Comment 15

The review brings up several studies, but it doesn’t touch on any limitations or potential biases in the cited research. Recognizing any gaps in the research or mentioning studies that have conflicting results would add a nice balance to the review.

Response

In response to your valuable suggestion, we would like to respectfully explain that while we fully acknowledge the importance of critically examining research limitations, the current state of this particular field presents some constraints. The available body of research on genetic markers of porcine aggression, while growing, still remains relatively limited in scope. Many of the studies represent pioneering work in this area, and as such, there are currently few direct replications or contradictory findings that would allow for a more comprehensive analysis of potential biases

Comment 16

Throughout the paper, there are some repetitive phrases popping up, like the frequent use of "furthermore," "in addition," and "this gene" in proximity. Streamlining this language could really enhance readability and cut down on redundancy.

Response

We have rephrased several sentences to reduce the frequency of terms such as "furthermore," "in addition," and "this gene."

Comment 17

The conclusion feels a bit too general and doesn't dive into specifics about the challenges and potential solutions. It mentions "genetic, environmental, and economic factors" but doesn't elaborate on what these are, which might leave readers scratching their heads. Including examples of these factors would really help anchor the conclusion in the specific context of the paper.

Response

We have included these points to conclusion section. (Lines 714-717).

Comment 18

While the conclusion does mention challenges, it doesn't really dig into them. For example, how do these genetic, environmental, and economic factors affect the practical implementation of strategies? A little more detail on the specific obstacles that need to be tackled could make the conclusion feel more grounded in the realities of the industry.

Response

Thank you for your comment. The detailed discussion of how genetic, environmental, and economic factors influence practical implementation is already covered in Sections 4-6 of the review. The Conclusion section was designed only to summarize these key points.

Comment 19

The term "new technologies" is vague. It would be much more impactful to mention specific technologies, like CRISPR gene editing, genomic selection, or other innovations, and explain how these tools could be used to address aggression.

Response

In this review, our primary goal was to provide a comprehensive overview of the current genetic, physiological, and environmental factors influencing aggressive behavior in pigs, rather than focusing on emerging genome-editing tools like CRISPR.

While we acknowledge the transformative potential of such technologies (e.g., CRISPR or genomic selection) for future research, this article specifically synthesizes existing knowledge on genetic markers (e.g., DRD2, SLC6A4), transcriptomic profiles, and practical challenges in breeding programs. We aimed to lay a foundation for understanding the complexity of aggression before exploring intervention strategies in depth.

That said, your suggestion is invaluable for future work! A follow-up review or perspective piece dedicated to applications of CRISPR, high-throughput genomic selection, or other innovations in mitigating aggression would indeed be a logical next step